# The anti-tumor activity of tangeretin in esophageal squamous cell carcinoma by inhibiting GLI2-mediated transcription of GPNMB

Dong Yang[1‡], Quan Zhang[1‡], Haoyong Kuang[2], Jian Liu[2], Sen Wu[1], Li Wei[1], Wenjian Yao[1]*

1 Department of Thoracic Surgery, Henan Provincial People's Hospital, People's Hospital of Zhengzhou University, School of Clinical Medicine, Henan University, Zhengzhou, Henan, China, 2 Department of Thoracic Surgery, Zhengzhou University People's Hospital, Henan Provincial People's Hospital, Zhengzhou, Henan, China

‡ DY and QZ are co-first authors on this work.
* doctor_yaowj@126.com

**Data Availability Statement:** All relevant data are within the paper and its Supporting Information files.

## Abstract

Tangeretin (Tan), a citrus flavonoid, possesses a strong anti-tumor efficacy in various human cancers. However, the precise role of Tan in the development of esophageal squamous cell carcinoma (ESCC) remains unclear. RNA sequencing (RNA-seq) analysis was performed to observe the Tan-related genes in Tan-treated TE-1 cells. The direct relationship between GLI family zinc finger 2 (GLI2) and the promoter of glycoprotein non-metastatic melanoma protein B (GPNMB) was predicted by bioinformatics analysis and validated by luciferase reporter and chromatin immunoprecipitation (ChIP) assays. Cell survival after Tan treatment was assessed by CCK8 assay. Gene expression levels were evaluated by a qRT-PCR, western blot, or immunofluorescence method. Cell migration and invasion were detected by wound-healing and transwell assays. The function of Tan *in vivo* was examined using xenograft studies. Our data indicated anti-migration and anti-invasion functions of Tan in ESCC cells *in vitro*. Tan also diminished tumor growth *in vivo*. Mechanistically, Tan diminished the expression and transcriptional activity of GLI2 in ESCC cells. Silencing of GLI2 resulted in decreased expression of GPNMB by inhibiting GPNMB transcription via the binding site at the GPNMB promoter at position +(1539–1550). Moreover, Tan down-regulated GPNMB expression in ESCC cells, and re-expression of GPNMB reversed anti-migration and anti-invasion functions of Tan in ESCC cells. Our findings uncover anti-migration and anti-invasion effects of Tan in ESCC cells by down-regulating GPNMB by suppressing GLI2-mediated GPNMB transcription, providing new evidence that Tan can function as a therapeutic agent against ESCC.

## Introduction

As the most frequent type of esophageal cancer, esophageal squamous cell carcinoma (ESCC) is a deadly and high-mortality malignancy worldwide [1, 2]. Although clinically therapeutic

**Funding:** This research was supported by Key Science and Technology Projects in Henan Province (No. 212102310669, No. 222102310045), and Henan Province Medical Science and Technology Research Plan Joint Construction Project (No. SBGJ202102022) and "23456 Talent Project" of Henan Provincial People's Hospital.

**Competing interests:** The authors have declared that no competing interests exist.

regimens, including surgery and radiochemotherapy, can treat many primary ESCC tumors effectively, these therapies remain restricted in controlling the metastasis of ESCC cells and curbing metastatic or advanced ESCC [3]. Hence, new anti-metastasis and anti-cancer therapeutic approaches are needed in ESCC.

Tangeretin (Tan), a type of bioactive polymethoxylated flavones found in citrus peel, has been unveiled to possess many pharmacological functions, such as anti-inflammatory, antioxidant, anti-viral, cardioprotective, neuroprotective, and hepatoprotective properties [4]. Of note, the anti-tumor efficacy of Tan has recently become increasingly clear due to its growth repression, apoptosis induction, anti-angiogenesis, and anti-metastasis effects [5]. Furthermore, Tan enhances the sensitivity of the chemotherapy drugs and reduces chemotherapy-induced toxicity [6, 7]. Functional studies in various cancer cell lines have highlighted the tumor-inhibitory potential of Tan [8–10], but its precise action in ESCC pathogenesis remains unclear.

Transcription factors (TFs) might prove crucial in promoting human carcinogenesis by controlling the transcription of the key genes by binding to their promoters [11]. GLI family zinc finger 2 (GLI2), a key TF that mediates the Sonic Hedgehog signaling, actively participates in human carcinogenesis [12, 13]. The activation and aberrant expression of GLI2 have been reported to induce the malignant phenotypes of multiple cancers, such as castration-resistant prostate cancer, renal cell carcinoma and gallbladder cancer [14–16]. In human esophagus adenocarcinoma, GLI2 is highly expressed and contributes to carcinoma formation [17]. Importantly, long noncoding RNA NR2F1-AS1 contributes to ESCC development by activating the Sonic Hedgehog pathway by elevating GLI2 expression [18], suggesting the oncogenic activity of GLI2 in ESCC. Although several other TFs, such as nuclear liver X receptor α (LXRα) and signal transducer and activator of transcription 3 (STAT3), have been illuminated as the downstream effectors of Tan [19, 20], no reports showed the involvement of GLI2 in the anti-cancer property of Tan in human tumors.

Here, we sought to elucidate the activity and molecular determinant of Tan in ESCC, with the hope that these findings might provide new evidence for developing Tan as a promising anti-tumor agent.

## Materials and methods

### Cell culture and treatment

Human ESCC cell line TE-1 (Procell, Wuhan, China) was propagated in 10% FBS RPMI-1640 (Servicebio, Wuhan, China), and the KYSE150 cell line (Procell) was cultured in 10% FBS RPMI-1640 containing Ham's F-12 (Servicebio). The 293T cell line (Procell) used for luciferase reporter assay was grown in complete medium provided by Procell. Human normal esophageal HET-1A cells (Wanwusw, Hefei, China) were cultured in BEGM Kit (Wanwusw). All cells were maintained in a fully humidified incubator in the presence of 5% $CO_2$ at a temperature of 37°C.

Tan (≥98% purity) was procured from Macklin (Shanghai, China) and dissolved in DMSO for all experiments. For treatment of Tan, TE-1 and KYSE150 cells were incubated with or without Tan for 24 h at a concentration of 20 μg/ml.

### Determination of cell survival after Tan treatment

The survival rate of Tan-treated cells was evaluated by CCK8 assay based on the standard protocols [21]. Cells ($5.0 \times 10^3$ cells/well) were seeded in regular growth media into 96-well culture dishes. 12 h post-seeding, the cells were challenged with different concentrations of Tan. After incubation of 48 h, 10 μl CCK8 reagent (Beyotime Biotechnology, Shanghai, China) was added

to each well. After addition, one-hour incubation was allowed at 37°C. Using a microplate reader (Thermo Fisher Scientific, MA, USA), the absorption was gauged at a wavelength of 450 nm. Surviving fraction (%) in Tan treatment group was estimated relative to that of the untreated counterpart. The IC50 value for Tan was determined from a plot of the percentage of surviving cells (50%) versus Tan concentration.

### RNA sequencing

For RNA sequencing (RNA-seq) as described [22], TE-1 cells were treated with 20 μg/ml Tan or DMSO for 24 h. RNA was prepared from the cells using magnetic beads with Oligo (dT) and fragmented. The quality and quantity of extracted RNA were analyzed using Qubit 2.0 fluorimeter (Thermo Fisher Scientific). RNA-seq analysis of high-quantity RNA for the construction of sequencing libraries was conducted by BGI Technology (Shenzhen, China) using the BGISEQ-500 sequencing platform (BGI Technology) following standard protocols. Read alignment mapped to the human genome grch37 was performed using Hisat2 software [23], and the read counts per group were calculated using the Python package HTseq [24]. The tab-delimited text files included FPKM values for each sample. Differentially expressed genes were defined under the conditions: log2|fold-change| > 1 and $P < 0.05$. Gene Ontology (GO) and Kyoto Encyclopedia of Genes and Genomes (KEGG) enrichment analyses were conducted using the R package "Cluster Profiler". A loop graph was drawn for the visualization of the enrichment results.

### Bioinformatics analysis

The differentially expressed genes (log2|fold-change| > 1 and $P < 0.05$) in ESCC tissues (n = 77) and normal esophageal tissues (n = 11) were obtained from The Cancer Genome Atlas (TCGA) database (https://www.tcga.org/). All human transcriptional factors (hTFs) were downloaded from AnimalTFDB database (http://bioinfo.life.hust.edu.cn/AnimalTFDB/). The TF binding sites for human were predicted using the Gene Transcription Regulation Database (GTRD) database (http://gtrd20-06.biouml.org/). To obtain the downstream targets of GLI2 TF, we used the database hTF-target at http://bioinfo.life.hust.edu.cn/hTFtarget#!/. To search the binding sites between GLI2 and the promoter of glycoprotein non-metastatic melanoma protein B (GPNMB), we interrogated the JASPAR database (https://jaspar.genereg.net/).

### siRNAs, plasmids and cell transfection

Human GLI2-specific siRNA (si-GLI2) and a non-silencing siRNA mock (si-NC) were procured from Wuzhoukangjian Biological Technology Co., LTD (Tianjin, China). The coding sequence of human GPNMB was obtained by PCR amplification and inserted into pECMV-MCS-FLAG expressing plasmid (Miaoling Biotechnology, Wuhan, China). As control, the scrambled sequence was processed in the same way to generate a plasmid control (vec).

TE-1 and KYSE150 cells ($5.0 \times 10^5$ cells/well) were seeded in 24-well culture dishes 24 h prior to transfection using Lipofectamine 3000 (Thermo Fisher Scientific). The following day, 50 nM of siRNA or/and 500 ng of plasmid was transfected into cells as per the manufacturing recommendations for transfection. Cells were collected 24 h post-transfection and subjected to Tan treatment.

### qRT-PCR of mRNA in ESCC cell lines

RNA preparation from TE-1 and KYSE150 cells after Tan treatment or/and transfection was done using RNAeasy™ Animal RNA Isolation Kit with Spin Column based on the

**Table 1. Primers sequences used for qRT-PCR.**

| Name | | Primers for qRT-PCR (5'-3') |
| --- | --- | --- |
| GLI2 | Forward | TTGACATGCGACACCAGGAA |
| | Reverse | AGAACGGAGGTAGTGCTCCA |
| BARX2 | Forward | CCAAGGAGACCTGCGATTACT |
| | Reverse | TGTTCCGTCTCTGACTCGCT |
| EN1 | Forward | GCAACCCGGCTATCCTACTT |
| | Reverse | TCGTTGAGGCTGAGTTCCTG |
| BMP7 | Forward | TGGTCCACTTCATCAACCCG |
| | Reverse | CCGGACCACCATGTTTCTGT |
| GPNMB | Forward | GCGAGATCACCCAGAACACA |
| | Reverse | ACACCAAGAGGGAGATCACAG |
| β-actin | Forward | CTCGCCTTTGCCGATCC |
| | Reverse | GGGGTACTTCAGGGTGAGGA |
| GPNMB promoter +(343–354) | Forward | CAAGCAATCACGAGCACAGG |
| | Reverse | TGTGGTGCCTCCCTCTCTAT |

recommendations of the manufacturer (Beyotime Biotechnology). For mRNA analysis, cDNA was oligo(dT) primed using SweScript RT II First Strand cDNA Synthesis Kit (Servicebio). Generated cDNA was diluted 10 folds, and qRT-PCR was performed on ABI QuantStudio6 Felx Real-Time System (Thermo Fisher Scientific) using the primers listed in Table 1. mRNA expression data were obtained using the $2^{-\Delta\Delta Ct}$ method [25] after normalization with reference to expression of β-actin.

## Western blot

Lysates of TE-1 and KYSE150 cells after Tan treatment or siRNA transfection were conducted using RIPA lysis buffer (Beyotime Biotechnology) with protease and phosphatase inhibitor cocktail (Beyotime Biotechnology) for 20–30 min on ice. For western blot [16], approximately 35 μg of protein was loaded per lane and electrophoresed on 5–12% SDS-PAGE gels. After the resulting gels were blotted onto PVDF membranes (Millipore, Bedford, MA, USA), probing was conducted with primary antibodies (rabbit polyclonal) including GLI2 (18989-1-AP, 1:1000, Proteintech, Wuhan, China), GPNMB (GB111475, 1:1000, Servicebio), Vimentin (GB111308, 1:1000, Servicebio), E-Cadherin (E-Cad, GB11082, 1:1000, Servicebio), N-Cadherin (N-Cad, GB111273, 1:1000, Servicebio), β-Tubulin (GB11017, 1:1000, Servicebio), Slug (12129-1-AP, 1:2000, Proteintech), Snail (13099-1-AP, 1:1000, Proteintech), VEGF (19003-1-AP, 1:1500, Proteintech), and Cyclin D1 (26939-1-AP, 1:10000, Proteintech), and β-actin (GB11001, 1:1000, Servicebio) antibodies. Following incubation with secondary HRP-tagged anti-rabbit IgG antibody (GB23303, 1:3000, Servicebio), signals were developed with Ultra sensitive ECL Chemiluminescence Kit as described by the manufacturer (Servicebio).

## Luciferase reporter assay

To evaluate the impact of Tan treatment on GLI2 transcriptional activity, $5.0 \times 10^5$ TE-1 cells grown in 24-well culture dishes were co-transfected using Lipofectamine 3000 with 100 ng pRL-TK control plasmid, constitutively expressing *Renilla* luciferase (for normalization, Promega), and 400 ng GLI2-luc luciferase reporter plasmid (Yeasen, Shanghai, China). Cells were subjected to Tan treatment (20 μg/ml, 24 h) at 24 h after transfection and assayed for luciferase activities. Firefly luciferase activity was normalized to that of *Renilla* luciferase.

To determine the binding site between GLI2 and the GPNMB promoter, the fragments (about 100 bp) of the GPNMB promoter containing the +(343–354) (named BS-1) position sequence or +(1539–1550) (named BS-2) position sequence, synthesized by Qingke Biotechnology (Beijing, China), were subcloned into the 3'UTR of the firefly luciferase coding region into pGL3-basic luciferase plasmid (Miaoling Biotechnology) as described previously [26]. For luciferase assays, $5.0 \times 10^5$ 293T cells growth in 24-well culture dishes were co-transfected using Lipofectamine 8000 (Beyotime Biotechnology) with reporter construct, siRNA and pRL-TK control plasmid based on the manufacturer's suggestion. 48 h post-transfection, cells were lysed for luciferase activities and firefly luciferase activity was normalized to that of *Renilla* luciferase.

## Chromatin immunoprecipitation (ChIP) assay

For ChIP assay [26], the ChIP Kit was applied as recommended by the manufacturer (Beyotime Biotechnology). Briefly, $1.0 \times 10^7$ TE-1 cells were fixed with 1% formaldehyde (Macklin) for 10 min, lysed in SDS cell lysis buffer (Beyotime), and sonicated to fragment DNA length into 200–800 bp using the Ultrasonic Cell Disruptor (Lichen, Shanghai, China). IP complexes were immunoprecipitated with anti-GLI2 antibody (ab226390, Abcam, Cambridge, UK) or rabbit IgG control (10284-1-AP, Proteintech) overnight at 4°C. DNA was purified with DNA Product Purification Kit (Solarbio, Beijing, China) and amplified by qRT-PCR using specific primers (Table 1).

## Measuring cell migration by wound-healing assay

TE-1 and KYSE150 cells after Tan treatment or/and plasmid transfection were seeded in 6-well culture dishes in 2 ml total volume to yield a monolayer of 80–90% confluence following overnight incubation. A homogeneous wound track was created using the 200 µl of sterile pipette tips. After 24 h culture in regular media at 37°C, images were acquired and analyzed using Image J software (National Institutes of Health, Bethesda, USA). We scored cell migration ability by gauging cell motility distance [27].

## Transwell migration and invasion assays

TE-1 and KYSE150 cells after Tan treatment or/and plasmid transfection were resuspended in non-serum medium and placed in 24-transwell inserts (Corning, New York, USA) precoated without (for migration analysis, $4.0 \times 10^5$ cells/well) or with (for invasion analysis, $2.0 \times 10^6$ cells/well) 1 mg/ml Matrigel (BD Biosciences). Then, the inserts were placed in 24-wells containing 10% FBS growth medium. Following 24-h incubation at 37°C, the cells that passed the membranes to the undersurface were photographed and counted in 10 random microscopic fields as described elsewhere [27].

## Xenograft studies

Six-week-old female BALB/c athymic nude mice (n = 10, Beijing Vital River Laboratory Animal Technology Co., Ltd., Beijing, China) were used for evaluation of the tumor-inhibitory effect of Tan. For formation of xenograft tumors [28], KYSE150 cells ($5 \times 10^6$ cells/mouse) were implanted into the right flanks of nude mice via subcutaneous injection. Five days later, the mice were administrated with Tan (20 mg/kg) or PBS at the same volume by intraperitoneal injection every three days. Each group included five mice. All mice were euthanized at day 35 by a $CO_2$ overdose method by gradually elevating the concentration of $CO_2$ in the anesthesia chamber. After the mice stopped breathing and heartbeat completely, the tumors were

removed, weighed, and paraffin-embedded. All animal procedures were conducted in accordance with a protocol approved by Animal Care and Use Committee of Henan Provincial People's Hospital.

## Immunofluorescence

For cell immunofluorescence staining [29], TE-1 and KYSE150 cells after Tan treatment or/ and plasmid transfection were fixed for 15 min in 4% paraformaldehyde, permeabilized by incubation for 20 min in 0.5% Triton X-100, and blocked for 30 min in 3% BSA at room temperature. For tumor immunofluorescence staining, paraffin-embedded tumors were sectioned at 4 μm thickness, dehydrated, boiled in 10 nM sodium citrate for 30 min, and blocked in 3% BSA for 30 min. PCNA or MMP9 was probed with a rabbit polyclonal antibody (Servicebio, Wuhan, China) to PCNA (GB11010, 1:500) or MMP9 (GB11132, 1:1000), respectively. The Alexa Fluor 488 conjugated goat anti-rabbit IgG (for cell staining, GB25303, 1:500, Servicebio) or Cy3 conjugated goat anti-rabbit IgG (for tumor staining, GB21303, 1:300, Servicebio) was used for visualization. Nuclei were subsequently stained by DAPI (Servicebio). Fluorescent images were obtained under an Olympus fluorescence microscope (BX53, Olympus, Tokyo, Japan) and analyzed using ImageJ software (NIH, Bethesda, MA, USA).

## Statistical analysis

Results were compared using ANOVA with Tukey's post hoc test (for multiple comparisons) or Student's $t$-test (two-tailed, for two-group comparison). Data were shown as mean ± SEM of three independent experiments ($n \geq 3$) that were done in triplicate. Difference between a single or combination treatment and control was considered significant at a $p$ value less than 0.05.

## Results

### Tan exerts anti-migration and anti-invasion functions in ESCC cells *in vitro*

We first evaluated the anti-tumor effects of Tan in TE-1 and KYSE150 ESCC cells *in vitro*. The cells were exposed to various concentrations of Tan, and cell viability was analyzed at 48 h. We found that exposure of Tan caused a remarkable reduction in cell survival rate, indicating that Tan hindered cell growth with the IC50 being 85.92 μg/ml in TE-1 cells and 94.34 μg/ml in KYSE150 cells (Fig 1A). We then treated cells with 20 μg/ml of Tan and evaluated the effect on cell migration and invasiveness after 24 h. Through wound-healing and transwell assays, we observed that treatment of Tan led to a striking reduction in the migratory abilities of the two ESCC cell lines (Fig 1B and 1C). Furthermore, treatment of Tan diminished cell invasiveness *in vitro* (Fig 1D). Immunofluorescence results showed that TE-1 and KYSE150 cells treated by Tan exhibited lower levels of proliferating marker PCNA and metastasis-related protein MMP9 compared with controls (Fig 2A and 2B). These results demonstrate that Tan can diminish ESCC cell growth, migration, and invasiveness, thereby exerting anti-ESCC activity.

### Tan diminishes tumor growth *in vivo*

To elucidate whether Tan possesses a tumor-inhibitory function, we implanted KYSE150 cells subcutaneously into the right flanks of nude mice, with administration of Tan every three days. As shown in Fig 3A and 3B, images showed a significant reduction in tumor volume as a result of Tan treatment. Moreover, Tan treatment reduced average weight of the xenograft tumors (Fig 3C). This result was also verified by immunofluorescence for cell proliferation

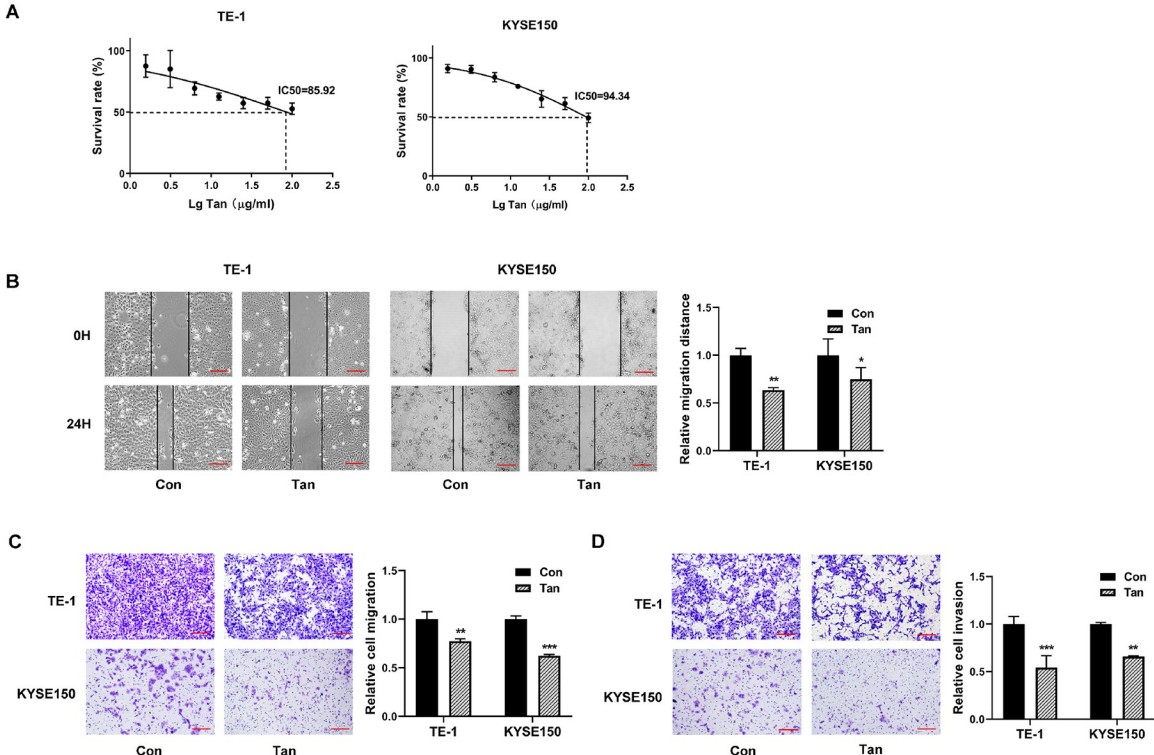

**Fig 1. Anti-migration and anti-invasion effects of Tan in TE-1 and KYSE150 ESCC cells.** (A) TE-1 and KYSE150 cells were exposed to incremental concentrations of Tan for 48 h and checked for cell viability by CCK8 assay. Also shown were the IC50 values in the two cell lines. (B) Cells were treated with 20 μg/ml of Tan for 24 h, and cell migration was quantified as the wound-healed distance. Also shown were the representative pictures. Scale bar: 100 μm. (C and D) Representative images depicting transwell migration and invasion assays performed with control (Con) or Tan-treated cells. Scale bar: 100 μm. *$P < 0.05$, **$P < 0.01$, ***$P < 0.001$.

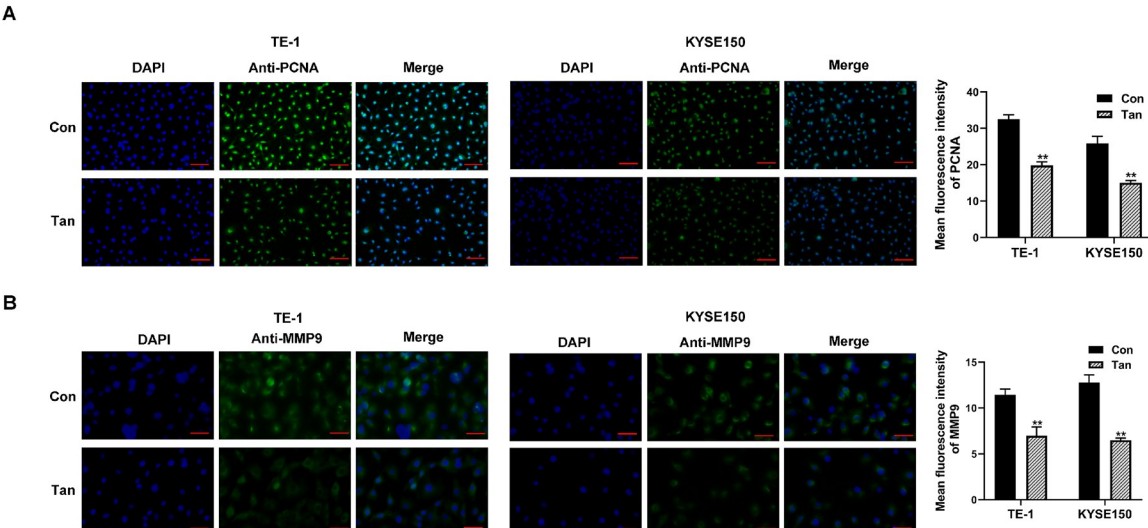

**Fig 2. Regulation of Tan in PCNA and MMP9 expression in ESCC cells.** (A and B) Immunofluorescence assay showing the fluorescence intensity of PCNA and MMP9 in cells treated with control (Con) or Tan. Scale bars: 100 μm (A) and 50 μm (B). **$P < 0.01$.

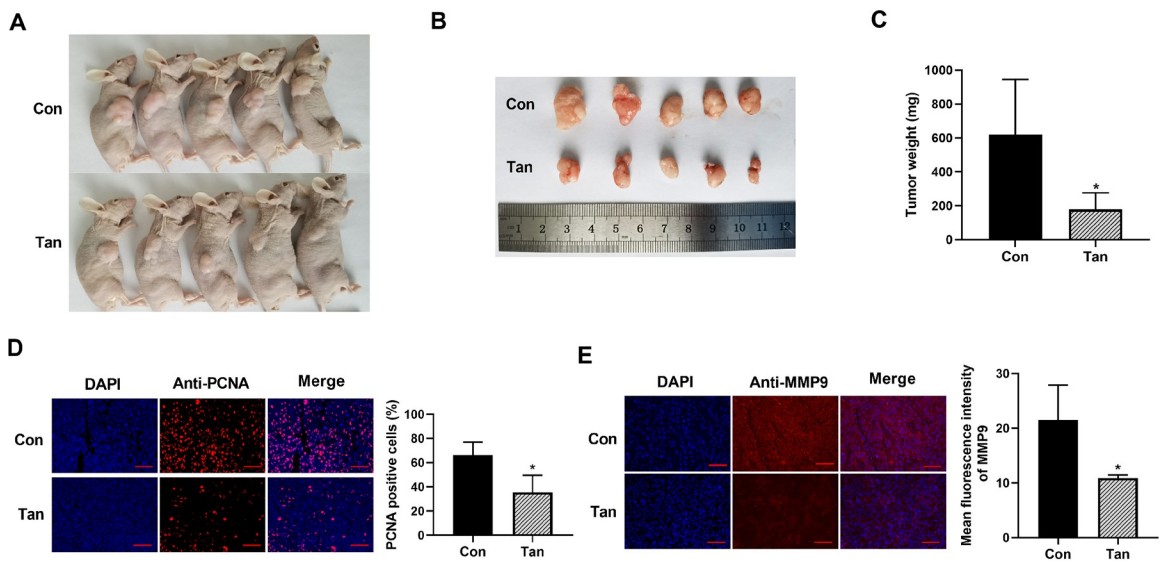

**Fig 3. The tumor-inhibitory effect of Tan *in vivo*.** KYSE150 cells ($5 \times 10^6$) were implanted subcutaneously into right flanks of nude mice, with administration of Tan every three days. 35 days later, the tumors were harvested from the mice. (A) Images of mice with the xenograft tumors. (B) Images of the xenograft tumors. (C) Mean weight of the xenograft tumors. (D and E) Immunofluorescence for PCNA and MMP9 levels in the harvested tumors. Scale bar: 100 μm. $^*P < 0.05$.

using PCNA. Tan treatment suppressed the expression of PCNA in the xenograft tumors (Fig 3D). Additionally, immunofluorescence results revealed that Tan treatment led to a clear down-regulation of MMP9 expression in the tumors (Fig 3E). All these data indicate the tumor-inhibitory effect of Tan *in vivo*.

## Transcriptional profiling of Tan-treated ESCC cells and selection of the regulatory TFs of Tan in ESCC

To find important genes in relation to the anti-ESCC activity of Tan, we performed RNA sequencing (RNA-seq) on Tan-treated TE-1 ESCC cells and control cells. We identified 587 genes with a significant variation following Tan treatment in TE-1 cells, in which 361 genes were down-regulated and 226 genes were up-regulated (S1 Table), and the cluster heat map of differentially expressed genes was shown in Fig 4A. We then performed GO and KEGG enrichment analyses to elucidate the mechanisms of the 587 genes on biological process and KEGG pathway. Results showed that these genes were closely associated with extracellular structure organization (GO: 0043062), collagen-containing extracellular matrix (GO: 0062023), extracellular matrix organization (GO: 0030198), ossification (GO: 0001503), heparin binding (GO: 008201), glycosaminoglycan binding (GO: 00055393), sulfur compound binding (GO: 1901681), extracellular matrix part (0044420), and interstitial matrix (0005614). KEGG pathway enrichment analysis pointed to the hippo signaling pathway (hsa04390), Complement and coagulation cascades (hsa04610), Basal cell carcinoma (hsa05217), wnt signaling pathway, and cholesterol metabolism (Fig 4B). To obtain the TFs that are not only aberrantly expressed in ESCC but also are controlled by Tan treatment in ESCC cells, we combined these genes including the 587 genes altered by Tan treatment in TE-1 ESCC cells (S1 Table), the all 1665 hTFs from AnimalTFDB database (S2 Table), and the differentially expressed genes in ESCC tissues and normal esophageal tissues using the TCGA database (S3 Table). As shown in Fig 4C, Venn diagram revealed a total of 18 TFs that could be regulated by Tan in TE-1 ESCC cells and were deregulated in ESCC. By combining the 18 TFs with the 2684 up-regulated

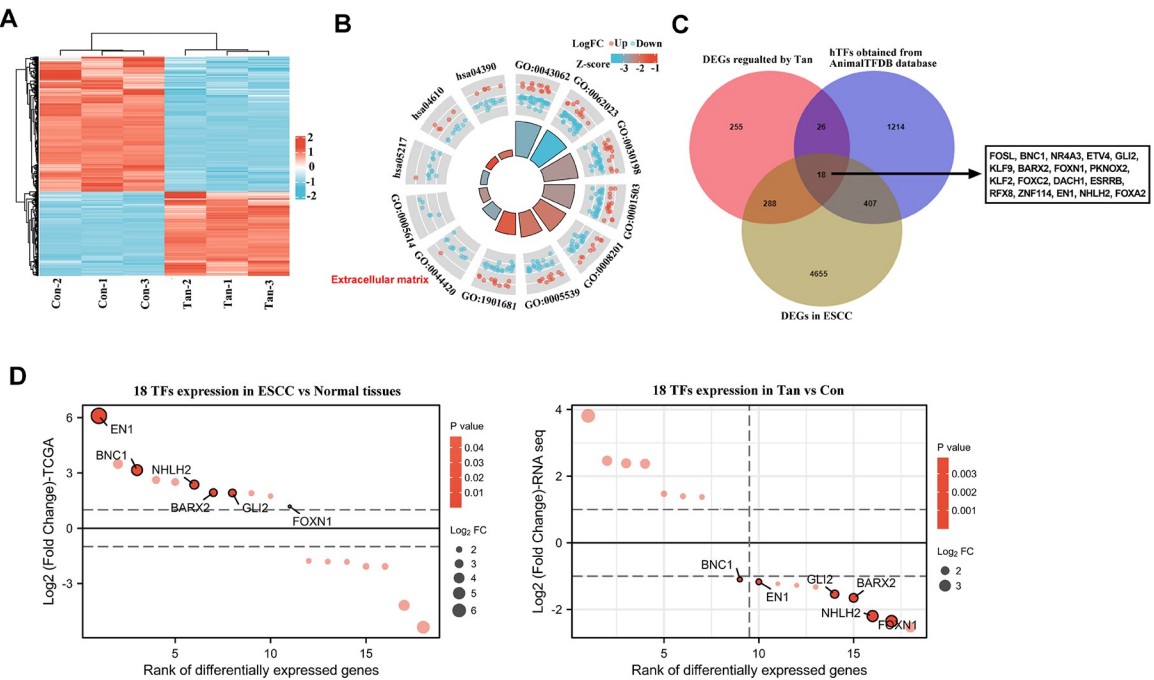

**Fig 4. Selection of TFs that were up-regulated in ESCC and were inhibited by Tan treatment in TE-1 ESCC cells.** (A) Heat map obtained from high-throughput RNA-seq profiling of Tan-treated TE-1 ESCC cells (20 μg/ml, 24 h) and control cells (Con). (B) The loop graph showing the markedly enriched biological processes, molecular function, cellular components, and KEGG pathways. The red and blue dots represent up-regulated and down-regulated genes, respectively. (C) Venn diagram revealing the 18 TFs that were not only aberrantly expressed in ESCC but also were controlled by Tan treatment in TE-1 ESCC cells. (D) Differential ordering diagram of 18 TFs in ESCC vs normal tissues, as well as in Tan treatment and con group.

genes in ESCC and the 361 genes down-regulated by Tan in ESCC TE-1 cells, we found a total of 6 TFs (BNC1, GLI2, BARX2, FOXN1, EN1 and NHLH2) (Fig 4D). Using the GTRD database, among the 6 TFs, we found that GLI2, BARX2 and EN1 had been shown to contain binding sites for human, which were identified by ChIP-seq experiments (-1000, +100).

## Tan reduces the expression and transcriptional activity of GLI2 in ESCC cells

Next, we wanted to evaluate whether the three TFs (GLI2, BARX2 and EN1) are involved in the anti-ESCC activity of Tan. We analyzed their expression in Tan-treated TE-1 and KYSE150 ESCC cells by qRT-PCR. Treatment of Tan led to a significant down-regulation in GLI2 mRNA expression, and BARX2 and EN1 mRNA levels did not reduce following Tan treatment in the cells (Fig 5A and 5B). We thus focused on the GLI2 TF as a potential target gene of Tan. Our western blot results further confirmed the reduction of GLI2 expression at the protein level following Tan treatment in TE-1 and KYSE150 ESCC cells (Fig 5C). To determine whether Tan could influence the transcriptional activity of GLI2, we adopted luciferase assays in TE-1 cells. Treatment of Tan caused a clear down-regulation in the luciferase activity of the GLI2-luc luciferase reporter plasmid (Fig 5D), supporting our hypothesis that Tan could inhibit the transcriptional activity of GLI2 in ESCC cells.

## Tan down-regulates GPNMB expression by suppressing GLI2-mediated transcription of GPNMB in ESCC cells

To identify GLI2 downstream genes responsible for the anti-ESCC activity of Tan, we integrated these genes including the 587 genes with a significant variation following Tan treatment

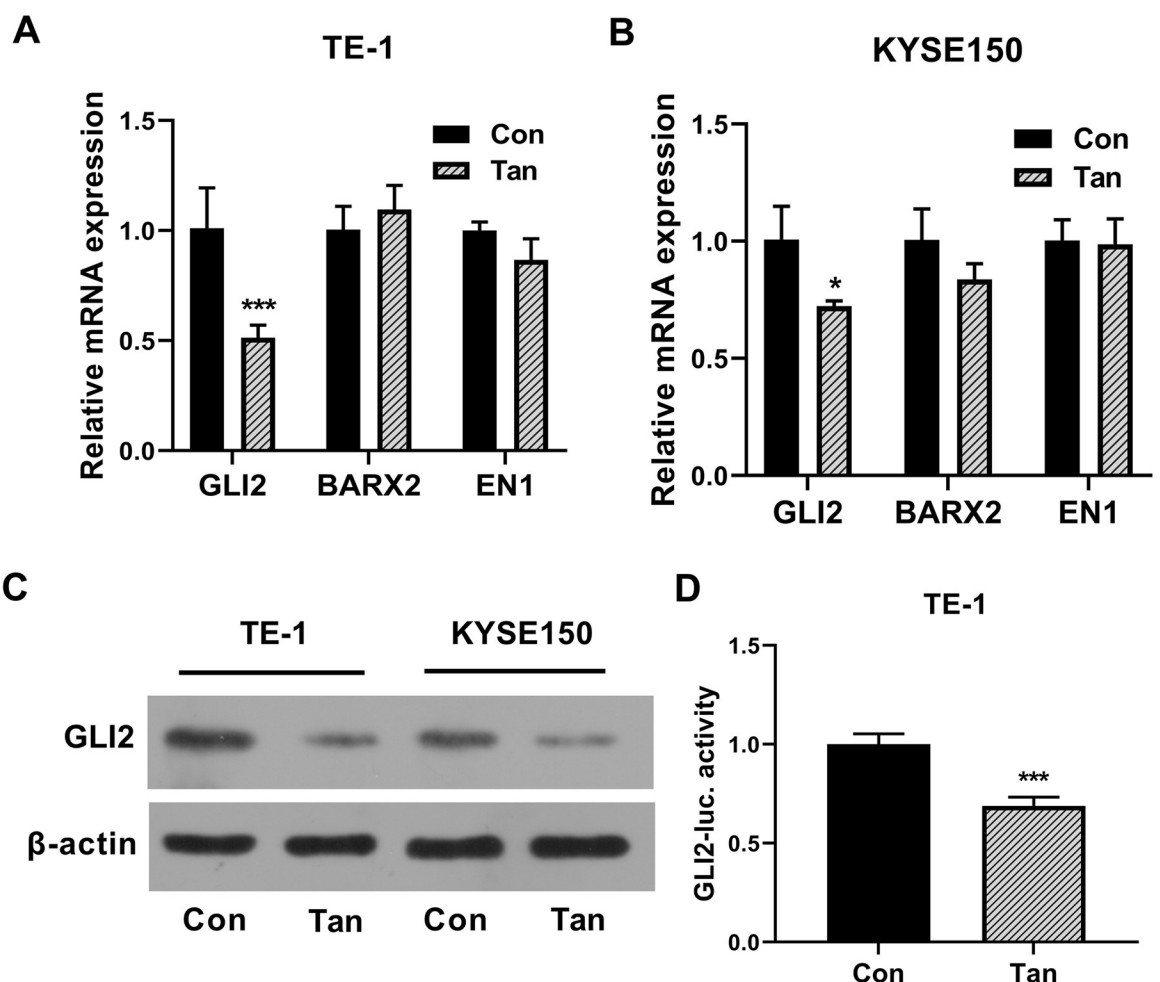

**Fig 5. Tan inhibits the expression and transcriptional activity of GLI2 in ESCC cells.** (A and B) qRT-PCR of the mRNA levels of GLI2, BARX2 and EN1 in Tan-treated TE-1 and KYSE150 ESCC cells (20 μg/ml, 24 h) and control cells (Con). (C) Western blot analysis of GLI2 protein expression in Tan-treated TE-1 and KYSE150 ESCC cells (20 μg/ml, 24 h) and control cells (Con). (D) TE-1 cells were introduced with GLI2-luc luciferase reporter plasmid and pRL-TK *Renilla* control vector and then subjected to Tan treatment (20 μg/ml, 24 h) or control (Con), followed by the evaluation of the luciferase activity. *$P < 0.05$, ***$P < 0.001$.

in TE-1 cells (S1 Table), the differentially expressed genes in ESCC tissues and normal esophageal tissues using the TCGA database (S3 Table), and the downstream target genes of GLI2 using hTF-target database (S4 Table). Of the 24 genes that overlapped among the 3 lists (Fig 5A), we found that 8 genes (ABCC3, BMP7, TM4SF1, GPNMB, COBLL1, FRY, BCAT1, ENHO) were positively correlated with GLI2 expression in ESCC tissues (Fig 6A). Furthermore, only GPNMB, BMP7, BCAT1 levels were markedly elevated in ESCC (Fig 6B, S3 Table) and were down-regulated following Tan treatment in TE-1 ESCC cells (Fig 6B, S1 Table). In consideration of the markedly correlation of GPNMB and BMP7 with $P < 0.01$ (Fig 6A), we further confirmed the expression of GPNMB and BMP7 in GLI2-lossed TE-1 and KYSE150 cells. As expected, silencing of GLI2 resulted in reduced mRNA levels of GPNMB in the two cell lines (Fig 6C), thereby, we selected GPNMB for further exploration. Western blot results also confirmed the down-regulation of GPNMB protein level in GLI2-silenced TE-1 and KYSE150 cells (Fig 6D), indicating that GLI2 positively regulated GPNMB expression in ESCC cells.

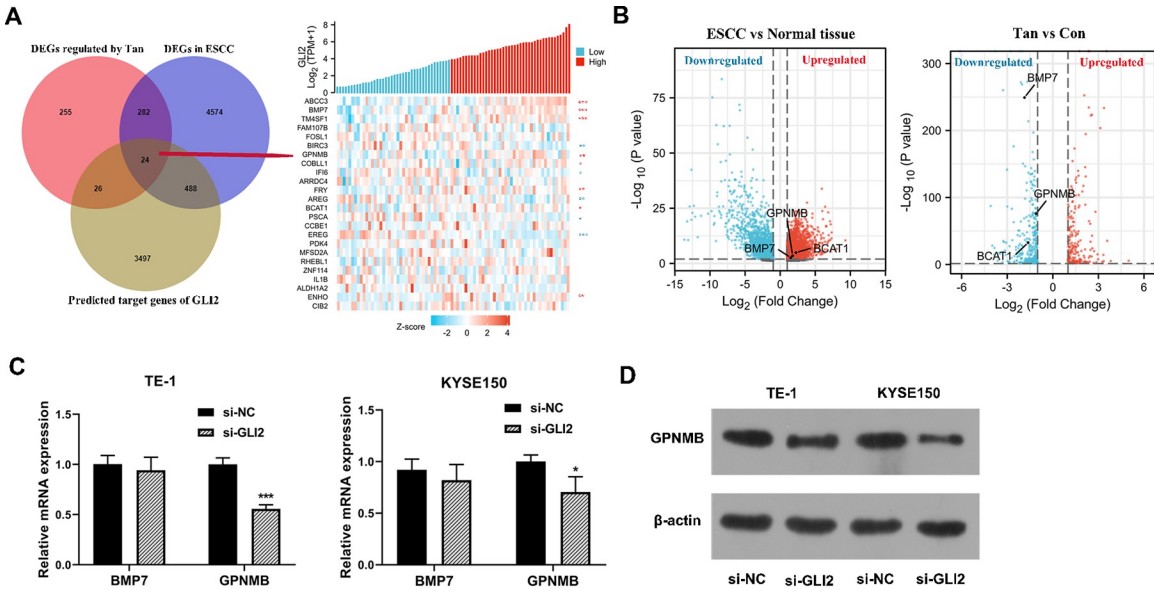

**Fig 6. GPNMB is a potential target gene of GLI2 in ESCC cells.** (A and B) Venn diagram revealing the 24 genes that overlapped among the 3 lists. Correlation clustering heat map of GLI2 and 24 target genes. Red "*" indicates positive correlation, and blue "*" indicates negative. (B) The volcano map of differentially expressed genes in ESCC vs normal tissue and Tan treatment vs Con. (C) qRT-PCR of the mRNA levels of GPNMB and BMP7 in TE-1 and KYSE150 ESCC cells after transfection by si-GLI2 or si-NC. (D) Western blot of GPNMB protein level in cells transfected as indicated. *$P < 0.05$, **$P < 0.01$, ***$P < 0.001$.

To search the binding region between GLI2 and the GPNMB promoter, we interrogated the JASPAR database and found a putative binding motif (Fig 7A) and two putative binding sites at positions +(343–354) (named BS-1) and +(1539–1550) (named BS-2) in the GPNMB promoter (Fig 7B). To verify this, we cloned the fragments (about 100 bp) of the GPNMB

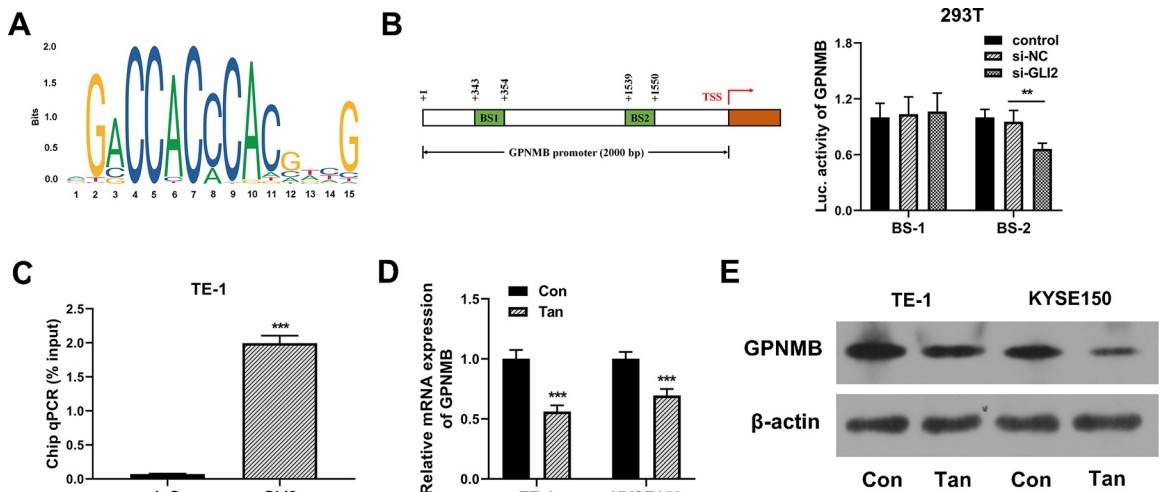

**Fig 7. Tan inhibits GPNMB expression via GLI2-mediated transcription of GPNMB in ESCC cells.** (A) The putative binding motif between GLI2 and the GPNMB promoter predicted by the JASPAR database. (B) Luciferase reporter activity from 293T cells cotransfected with the BS-2 reporter construct or the BS-1 reporter plasmid and si-GLI2 or si-NC. (C) ChIP assay with the lysates of TE-1 cells using an antibody against GLI2 or IgG control. qPCR with specific primers was utilized to gauge the enrichment level of the GPNMB promoter at position +(1539–1550). (D and E) GPNMB mRNA level by qRT-PCR and its protein expression by western blot in Tan-treated TE-1 and KYSE150 ESCC cells (20 μg/ml, 24 h) and control cells (Con). **$P < 0.01$, ***$P < 0.001$.

promoter containing the +(343–354) (named BS-1) position sequence or +(1539–1550) (named BS-2) position sequence into the pGL3-basic luciferase plasmid and transfected them into 293T cells together with si-GLI2. With the BS-2 reporter construct and GLI2 silencing led to a reduction in luciferase activity (Fig 7B). When the BS-1 reporter plasmid was introduced, no reduction in luciferase was observed with GLI2 silencing (Fig 7B), demonstrating that GLI2 could bind to the GPNMB promoter through the site at position +(1539–1550). Furthermore, ChIP experiments showed that the GPNMB promoter at position +(1539–1550) was preferentially enriched in GLI2-associating immunoprecipitates compared with IgG controls (Fig 7C), reinforcing the binding relationship between GLI2 and the GPNMB promoter. All these findings indicate that GLI2 positively regulates GPNMB transcription by binding to the GPNMB promoter.

Having established that Tan inhibits GLI2 transcriptional activity, we next determined whether Tan could affect GPNMB expression in ESCC cells. Notably, treatment of Tan downregulated GPNMB expression at both mRNA and protein in TE-1 and KYSE150 ESCC cells (Fig 7D and 7E). In summary, these observations suggest that Tan inhibits GPNMB expression at least in part by suppressing GLI2-mediated transcription of GPNMB in ESCC cells.

## Re-expression of GPNMB reverses anti-migration and anti-invasion functions of Tan in ESCC cells

To determine whether the anti-tumor activity of Tan in ESCC is due to the reduction of GPNMB, a GPNMB cDNA plasmid was introduced before Tan treatment and assayed for cell migration and invasiveness. The efficacy of the cDNA plasmid in increasing GPNMB expression was validated by qRT-PCR in ESCC cells (Fig 8A). Indeed, re-expression of GPNMB strongly abolished Tan-driven suppression of viability, migration, and invasiveness of TE-1 and KYSE150 cells (Fig 8B–8E). Furthermore, GPNMB re-expression abated Tan-mediated reduction of PCNA and MMP9 levels in the two cell lines (Fig 9A and 9B). Collectively, these findings point to the notion that Tan diminishes ESCC cell migration and invasion partially via the suppression of GPNMB.

## Discussion

Emerging evidence supports the notion that Tan exhibits potent anti-tumor efficacy against numerous human cancers [5, 6]. Identifying the molecular determinants underlying the activity of Tan has been challenging. In the current work, we first demonstrate the anti-cancer activity of Tan in TE-1 and KYSE150 ESCC cells. Furthermore, we uncover the relevance of GLI2-mediated GPNMB transcription to the anti-ESCC activity of Tan, providing a new understanding of the inhibitory mechanism of Tan in ESCC development and a rationale for developing Tan as a potential agent against ESCC.

Functional studies in cancer cell lines have unveiled the tumor-inhibitory potential of Tan. For instance, in hepatocellular carcinoma HepG2 cells, Tan possesses the suppressive effects on cell growth and migration by elevating autophagy via the JNK/Bcl-2/beclin1 pathway [30]. In colorectal carcinoma COLO205 cells, Tan strongly enhances cell cycle arrest at G1 phase at least in part by diminishing cyclin-dependent kinases 2 (Cdk2) and 4 (Cdk4) activities [31]. Treatment of Tan in prostate cancer PC-3 cells leads to suppressed EMT by attenuating Vimentin expression and increasing E-cad level through the PI3K/Akt/mTOR pathway [32]. Moreover, Tan-zinc oxide quantum dots perform a strong anti-metastasis property in metastatic lung cancer NCI-H358 cells [33]. In the current report, we identify, for the first time, the anti-migration, anti-invasion, and anti-tumor functions of Tan in ESCC cells. Intriguingly, Tan (1–100 μg/ml) did not significantly affect the growth of normal esophageal HET-1A cells,

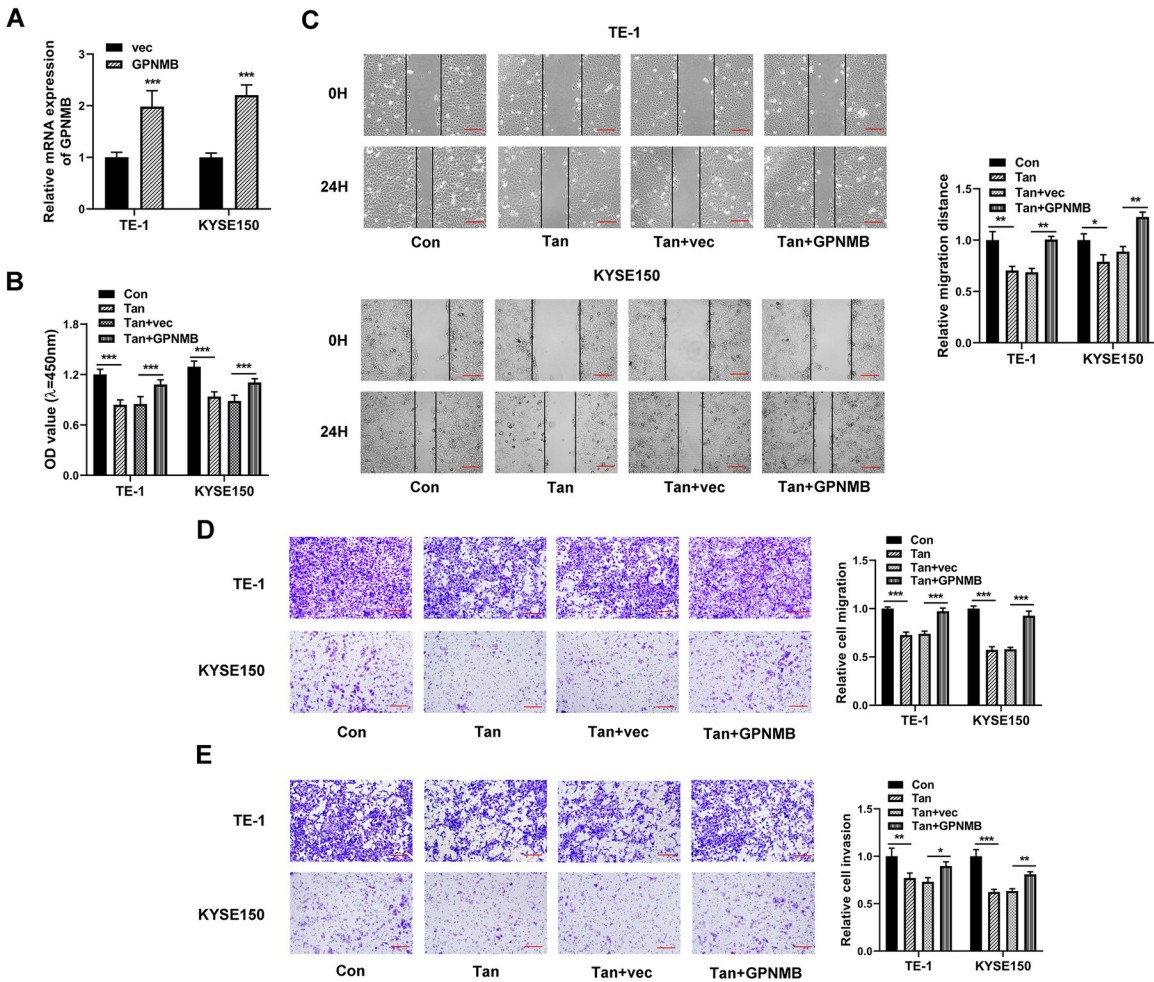

**Fig 8. Anti-migration and anti-invasion activities of Tan in ESCC cells are due to the down-regulation of GPNMB.** (A) qRT-PCR of GPNMB mRNA in cells transfected with vec or GPNMB. (B-E) TE-1 and KYSE150 cells were introduced with or without the GPNMB cDNA plasmid or control plasmid (vec) before Tan treatment (20 μg/ml, 24 h) and control treatment (Con). (B) Cell viability was determined by CCK8 assay. (C) Representative images showing a cell migration assay with cells treated as indicated and cell migration by wound-healing assay. Scale bars: 100 μm. (D and E) Representative images depicting transwell migration and invasion assays performed with cells treated as indicated. Scale bar: 100 μm. *$P < 0.05$, **$P < 0.01$, ***$P < 0.001$.

and the IC50 value of Tan was 520.6 μg/ml against HET-1A cells (S1 Fig), suggesting the safety of Tan at low doses as an anti-tumor drug.

By analyzing the transcriptional profiling of Tan-stimulated TE-1 ESCC cells, the GLI2 TF seems to be a candidate for the Tan target genes. The Sonic Hedgehog signaling has established a vital role in cancer biology [34]. As the downstream gene of the pathway, numerous reports have implicated GLI2 in the regulation of human carcinogenesis by inducing cancer cell malignant phenotypes, suggesting the potential of GLI2 as a new target for cancer treatment [13]. For example, GLI2 participates in the enhanced invasive ability of gallbladder cancer cells by increasing cell EMT [16]. Knocking down GLI2 in cervical cancer cells can cause suppressed cell growth and migration [35]. Moreover, GLI2 has been identified as a potent oncogene in esophageal cancer, including ESCC, and it is present at high levels in these diseases [17, 18]. Our results first discovered that in TE-1 and KYSE150 ESCC cells, Tan down-regulates GLI2 expression by diminishing the transcriptional activity of GLI2. We thus speculate that GLI2 deregulation might be responsible for the anti-ESCC activity of Tan. In addition to the

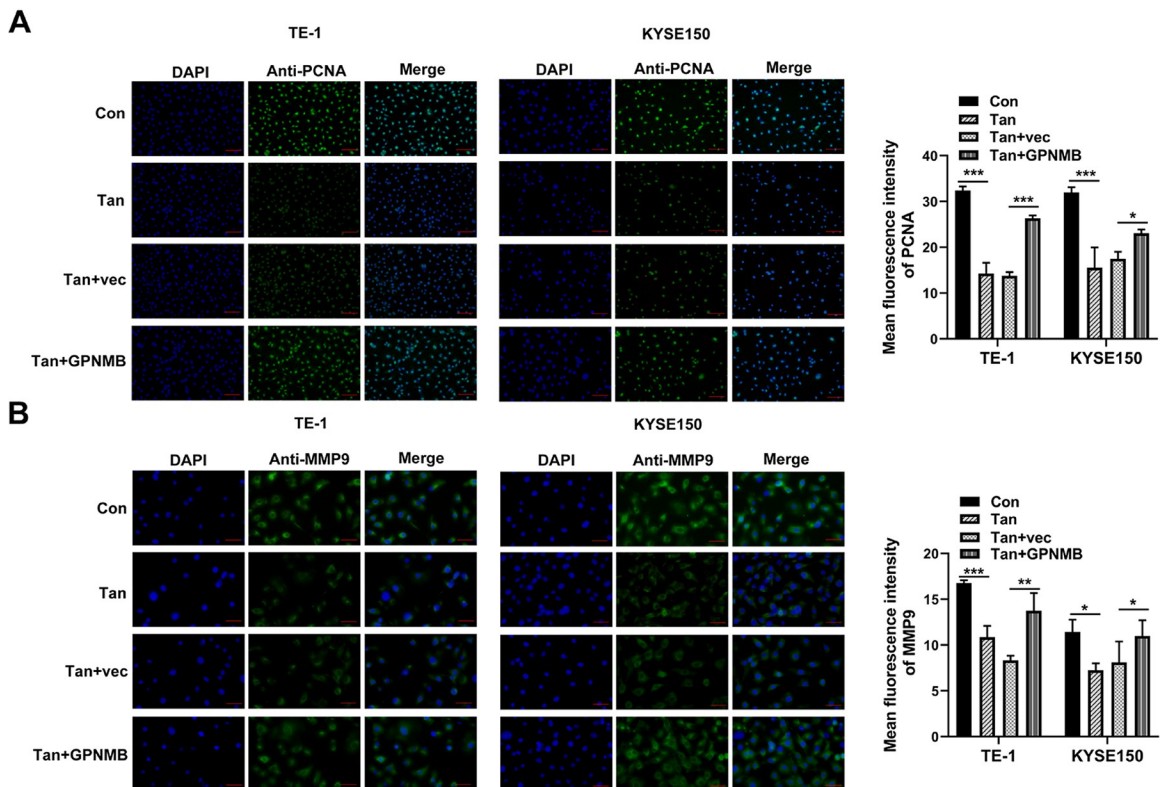

**Fig 9. Regulation of Tan in the expression of PCNA and MMP9 in ESCC cells through GPNMB.** (A and B) TE-1 and KYSE150 cells were introduced with or without the GPNMB cDNA plasmid or control plasmid (vec) before Tan treatment (20 μg/ml, 24 h) and control treatment (Con), followed by the detection of PCNA (A) and MMP9 (B) expression by immunofluorescence. Scale bars: 100 μm (A) and 50 μm (B). *$P < 0.05$, **$P < 0.01$, ***$P < 0.001$.

Hedgehog signaling, GLI2 has identified as a direct target of the TGF-β/SMAD pathway [13]. Previous work also uncovered the inhibition of Tan in TGF-β1 expression in human retinal pigment epithelial cells [36]. These conclusions suggest that Tan may reduce GLI2 expression by suppressing the TGF-β/SMAD pathway. Related field research is warranted in future work.

TFs can modulate the transcription of the key cancer-related genes and thus actively participate in human carcinogenesis [11]. According to a series of bioinformatics analyses and our siRNA experiments, GPNMB is identified as a target gene of GLI2. Furthermore, we first define that GLI2 positively regulates GPNMB transcription by binding to the GPNMB promoter at position +(1539–1550). GPNMB, an endogenous transmembrane protein, has emerged as a critical positive contributor to human carcinogenesis and is present at high levels in various types of cancers, thereby targeting GPNMB as an attractive strategy against cancer [37, 38]. GPNMB functions as a pro-tumorigenic factor predominantly by promoting tumor migration and invasiveness and elevating the levels of tumorigenic factors [37]. Bioinformatics analysis revealed the overexpression of GPNMB in ESCC. Nonetheless, no reports proved whether GPNMB overexpression is causally involved in ESCC pathogenesis. In the current work, we first demonstrate the inhibitory effect of Tan on GPNMB expression in TE-1 and KYSE150 ESCC cells. More intriguingly, our study defines the causal association between the reduction of GPNMB and the anti-migration and anti-invasion functions of Tan in TE-1 and KYSE150 ESCC cells. Some important metastasis-related markers, such as EMT specific molecule Snail, angiogenesis-related molecule VEGF, and cell cycle regulator Cyclin D1, have been

identified as target genes of GLI2 TF [39, 40]. Our data also uncovered the repression of Tan in the expression of Slug, Snail, VEGF, and Cyclin D1 in ESCC cells (S2 Fig), suggesting that Tan may downregulate their expression via GLI2 and thus influences ESCC metastasis. Future work will build on the findings by elucidating whether Tan exerts an anti-metastasis function in ESCC *in vivo* and whether the novel mechanism is involved in the anti-metastasis function of Tan *in vivo*. Moreover, more studies are required to determine the long-term efficacy and safety of Tan in various experimental models. Additionally, MMP9 is a crucial regulator implicated in ESCC outcome and prognosis [41], and its expression can be regulated by GLI2 in melanoma cells [42]. A previous document proved that GPNMB can influence MMP9 expression in chronic obstructive pulmonary disease [43]. Our data also showed that Tan can downregulate MMP9 expression by diminishing GPNMB level in ESCC cells. With these findings, we speculate that Tan in ESCC cells down-regulates MMP9 expression by suppressing GLI2--mediated GPNMB transcription.

Owing to the diverse pharmacological activities and minimal side effects, the flavonoids, including Tan, have gained much attention in the prevention and treatment of cancer [5]. However, certain limitations, such as the difficult purification process and poor solubility, impede their clinical use [5]. Therefore, it is very important to study how these limitations are resolved for their clinical use.

## Conclusions

Here, we conclude that Tan exerts anti-migration and anti-invasion effects in ESCC cells by down-regulating GPNMB by suppressing GLI2-mediated GPNMB transcription, providing a new understanding of the tumor-inhibitory mechanism of Tan and the basis for the development of Tan as a therapeutic agent against ESCC.

## Supporting information

**S1 Fig. The IC50 value of Tan in normal esophageal HET-1A cells.**
(TIF)

**S2 Fig. Effect of Tan on the expression of VEGF, Slug, Snail and Cyclin D1.**
(TIF)

**S1 Table. The 587 genes with a significant variation following Tan treatment in TE-1 cells.**
(XLSX)

**S2 Table. The all 1665 human transcriptional factors.**
(XLSX)

**S3 Table. The differentially expressed genes in ESCC tissues and normal esophageal tissues using the TCGA database.**
(XLSX)

**S4 Table. The genes that could be regulated by GLI2 from hTF-target database.**
(XLSX)

**S1 Raw images.**
(PDF)

## Author Contributions

**Conceptualization:** Wenjian Yao.

**Data curation:** Haoyong Kuang, Jian Liu.

**Formal analysis:** Jian Liu, Sen Wu.

**Investigation:** Dong Yang.

**Methodology:** Quan Zhang, Haoyong Kuang, Sen Wu, Li Wei.

**Project administration:** Dong Yang, Quan Zhang, Wenjian Yao.

**Resources:** Sen Wu.

**Software:** Haoyong Kuang, Jian Liu.

**Supervision:** Haoyong Kuang, Li Wei.

**Validation:** Dong Yang, Li Wei, Wenjian Yao.

**Visualization:** Wenjian Yao.

**Writing – original draft:** Quan Zhang.

**Writing – review & editing:** Wenjian Yao.

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
