## [Decision Letter · Decision Letter 0]

27 Feb 2023

PONE-D-23-03144The anti-tumor activity of tangeretin in esophageal squamous cell carcinoma by inhibiting GLI2-mediated transcription of GPNMBPLOS ONE

Dear Dr. Yao,

Thank you for submitting your manuscript to PLOS ONE. After careful consideration, we feel that it has merit but does not fully meet PLOS ONE’s publication criteria as it currently stands. Therefore, we invite you to submit a revised version of the manuscript that addresses the points raised during the review process. Your manuscript has been reviewed by three independent reviewers and all of them provided positive reviews. You will find their individual comments below and notice that many of them suggests minor revisions. Only one experiment has been suggested by reviewer#3. 

We look forward to receiving your revised manuscript.

Kind regards,

Arunava Roy, Ph.D.

Academic Editor

PLOS ONE

Journal Requirements:

Reviewers' comments:

Reviewer's Responses to Questions

**Comments to the Author**

1. Is the manuscript technically sound, and do the data support the conclusions?

Reviewer #1: Yes

Reviewer #2: Yes

Reviewer #3: Yes

2. Has the statistical analysis been performed appropriately and rigorously? 

Reviewer #1: Yes

Reviewer #2: Yes

Reviewer #3: Yes

3. Have the authors made all data underlying the findings in their manuscript fully available?

Reviewer #1: Yes

Reviewer #2: Yes

Reviewer #3: Yes

4. Is the manuscript presented in an intelligible fashion and written in standard English?

Reviewer #1: Yes

Reviewer #2: Yes

Reviewer #3: Yes

5. Review Comments to the Author

Reviewer #1: 1. Provide at least a reference for the given methodology mentioned in materials and methods.

2. In table 1, check the page format which affected the name EN1

3. Provide the scale bar for fig.1 & 7 B, arrange an original (un-edited) figure for the fig.1&7 B (Wound-healing assay) in supplementary info.

4. Figure 1 looks packed, it’s better to split into 2 figures for better visualization.

5. In the figure 1E and F, the resolution is poor to understand the differences between treated versus control. Increase resolution and mark the changes with arrow.

6. Provide the approval info in manuscript, regarding animal study and ethical concerns.

7. The figure 2D&E about Immunofluorescence for PCNA and MMP9, the finding are not clear. More resolution and highlight the difference.

8. Unable to read the legends of figure 3 and 5, please increase the resolution and font size.

Reviewer #2: In this manuscript, Quan Zhang et al., describes the anti-migration and anti-invasion properties of a flavonoid natural product tangeretin in esophageal squamous cell carcinoma (ESCC) cells. In xenograft tumor mouse model, tangeretin reduced the tumor growth and decreased the proliferation markers. Using gene silencing experiments, Chip assays, RNA seq and bioinformatic analysis, the authors have shown that suppression of GLI-2 mediated GPNMB transcription is responsible for the observed anticancer activity. Even though various publications highlight the potential anticancer properties of tangeretin, the results presented in this manuscript is interesting and the studies were well planned and executed. I would recommend this manuscript after addressing the following minor issues.

1. Scale bars should be included in figures or figure legends where fluorescent images were reported.

2. The author showed that tangeretin suppressed GLI-2 mediated GPNMB transcription in ESCC cells. Through invitro and invivo experiments the authors showed that tangeretin reduced the expression of MMP-9 levels. Is there a shared mechanism behind these two processes, or does GLI2 downregulation result in decreased MMP-9 expression in ESCC? For example, various articles (example. Chin J Cancer Res. 25(6), 637–645) explaining the role of MMP-9 in ESCC and other cancers. Especially, Faião-Flores et al., (Oncogene, 36, 2017, 1849–186) showed that downregulating GLI2 expression leads to reduced MMP-9 expression in melanoma cell lines. In line with these findings, I would recommend the authors to briefly discuss about this relationship in the manuscript.

3. Even though tangeretin acts by downregulating GLI2 transcription factor, the molecular target of tangeretin is still unknown. I would encourage the authors to briefly comment about the upstream activators of GLI2 as the possible molecular target of tangeretin.

4. Is there any weight loss noticed with the treatment group compared to the control group in vivo tumor xenograft studies?

5. In the abstract, background section, the following sentence should be rewritten,

“However, the precise action of Tan in the development of esophageal squamous cell carcinoma (ESCC) has remained a mystery”

6. The manuscript will get benefit from additional English proof reading.

Reviewer #3: Remarks to the Author:

The article “The anti-tumor activity of tangeretin in esophageal squamous cell carcinoma by

inhibiting GLI2-mediated transcription of GPNMB’’ is an interesting study detecting the effect of tangeretin on migration and invasion in esophageal cancer cells by downregulating GLI2-mediated transcription.

However, the existing data are partially supportive of the authors' conclusions. Additional experiments are needed to fully support their findings. Minor revision is needed in recommending this manuscript for publication.

Minor issues:

1. In Fig.1A the author mentioned that the IC50 of tangeretin in TE-1 and KYSE150 is 85.92 and 94.34 μg/ml respectively, but why the author has selected the dose of 20 μg/ml concentration for evaluation of the anti-migratory effect tangeretin in these cells is unclear. Moreover, the IC50 value of tangeretin in normal esophageal cells need to be included in the result. The effect of tangeretin on normal cell growth and proliferation need to be included.

2. Authors mentioned that Gli2 downregulation by tangeretin causes reduction in cancer cell migration and invasion. Authors only showed the expression of MMP9 and PCNA, but Gli2 itself controls the transcription of EMT (epithelial-mesenchymal transition) specific molecules like Slug, Snail and also controls molecules like VEGF, Cyclin D1, Cyclin D2, Cyclin E. Hence the effect of tangeretin on the expression these proteins should be determined by western blot analysis.

3. The resolution of Fig 3 (A-D), Fig. 5 A,B are very poor and should be changed with higher resolution figures.

6. PLOS authors have the option to publish the peer review history of their article (what does this mean?). If published, this will include your full peer review and any attached files.

Reviewer #1: No

Reviewer #2: No

Reviewer #3: No

---

## [Author Response · Author response to Decision Letter 0]

24 Jun 2023

Dear Editor,

We have revised the manuscript according to Reviewers’ comments and journal requirements. We look forward to hearing from you. Thank you very much.

Journal Requirements:

Response: We have revised the style and ensured that our manuscript meets PLOS ONE’s style requirements.

Response: We have added the details about the above questions in the Revised manuscript.

Response: We have provided raw images including all western blots and wound-healing assays in the Supporting Information.

Response: We have provided all raw images for western blot in the Supporting Information in the Revised manuscript.

Response: We have added the ORCID iD of the corresponding author in the Revised manuscript.

Response: We have added captions for the Supporting Information files at the end of our Revised manuscript.

Response: We have checked the reference list and ensured that no retracted documents were cited in our manuscript. We have also revised the reference style according to the PLOS ONE's style requirements in the Revised manuscript.

Reviewers' comments:

Reviewer's Responses to Questions

Comments to the Author

1. Is the manuscript technically sound, and do the data support the conclusions?

Reviewer #1: Yes

Reviewer #2: Yes

Reviewer #3: Yes

2. Has the statistical analysis been performed appropriately and rigorously?

Reviewer #1: Yes

Reviewer #2: Yes

Reviewer #3: Yes

3. Have the authors made all data underlying the findings in their manuscript fully available?

Reviewer #1: Yes

Reviewer #2: Yes

Reviewer #3: Yes

4. Is the manuscript presented in an intelligible fashion and written in standard English?

Reviewer #1: Yes

Reviewer #2: Yes

Reviewer #3: Yes

5. Review Comments to the Author

Reviewer #1:

1. Provide at least a reference for the given methodology mentioned in materials and methods.

Response: Thank you for your valuable suggestion. We have provided references for the methods in the Revised manuscript.

2. In table 1, check the page format which affected the name EN1

Response: Thank you. We have checked the page format, and revised it.

3. Provide the scale bar for fig.1 & 7 B, arrange an original (un-edited) figure for the fig.1&7 B (Wound-healing assay) in supplementary info.

Response: Thank you for your kind recommendations. We have added the scale bars in Revised Figure 1 and 7 and manuscript. We also provided the original figures for wound-healing assay in the Supporting information.

4. Figure 1 looks packed, it’s better to split into 2 figures for better visualization.

Response: We have split the Figure 1 into 2 figures in the Revised manuscript.

5. In the figure 1E and F, the resolution is poor to understand the differences between treated versus control. Increase resolution and mark the changes with arrow.

Response: Thank you for your valuable suggestion. We have increased resolution in the Revised Figure 2A and 2B. In these assays, Tan reduced the expression of PCNA and MMP9 in ESCC cells, as presented by the reduced mean green fluorescence intensity.

6. Provide the approval info in manuscript, regarding animal study and ethical concerns.

Response: We have provided the approval info in manuscript, regarding animal study and ethical concerns in the Revised manuscript as below.

All animal procedures were conducted in accordance with a protocol approved by Animal Care and Use Committee of Henan Provincial People's Hospital. (page 9, paragraph 2)

7. The figure 2D&E about Immunofluorescence for PCNA and MMP9, the finding are not clear. More resolution and highlight the difference.

Response: According to the comment, we have provided more resolution pictures in the Revised Figure 3D and 3E. In these assays, Tan reduced the expression of PCNA and MMP9 in xenograft tumors, as presented by the reduced PCNA positive cells (red-stained cells) and mean red fluorescence intensity of MMP9, respectively.

8. Unable to read the legends of figure 3 and 5, please increase the resolution and font size. 

Response: Thank you for your valuable suggestion. We have increased the solution and font size in the Revised Figure 4 and 6.

Reviewer #2:

In this manuscript, Quan Zhang et al., describes the anti-migration and anti-invasion properties of a flavonoid natural product tangeretin in esophageal squamous cell carcinoma (ESCC) cells. In xenograft tumor mouse model, tangeretin reduced the tumor growth and decreased the proliferation markers. Using gene silencing experiments, Chip assays, RNA seq and bioinformatic analysis, the authors have shown that suppression of GLI-2 mediated GPNMB transcription is responsible for the observed anticancer activity. Even though various publications highlight the potential anticancer properties of tangeretin, the results presented in this manuscript is interesting and the studies were well planned and executed. I would recommend this manuscript after addressing the following minor issues.

1. Scale bars should be included in figures or figure legends where fluorescent images were reported.

Response: Thank you for your valuable suggestion. We have added scale bars in Revised figures and figure legends.

2. The author showed that tangeretin suppressed GLI-2 mediated GPNMB transcription in ESCC cells. Through in vitro and in vivo experiments, the authors showed that tangeretin reduced the expression of MMP-9 levels. Is there a shared mechanism behind these two processes, or does GLI2 downregulation result in decreased MMP-9 expression in ESCC? For example, various articles (example. Chin J Cancer Res. 25(6), 637–645) explaining the role of MMP-9 in ESCC and other cancers. Especially, Faião-Flores et al., (Oncogene, 36, 2017, 1849–186) showed that downregulating GLI2 expression leads to reduced MMP-9 expression in melanoma cell lines. In line with these findings, I would recommend the authors to briefly discuss about this relationship in the manuscript.

Response: Thank you for your valuable suggestion. We have discussed the related relationship in the Revised manuscript as below.

Additionally, MMP9 is a crucial regulator implicated in ESCC outcome and prognosis [41], and its expression can be regulated by GLI2 in melanoma cells [42]. A previous document proved the modulation of GPNMB in MMP9 in chronic obstructive pulmonary disease [43]. Our data also showed that Tan can down-regulate MMP9 expression by diminishing GPNMB level in ESCC cells. With these findings, we speculate that Tan in ESCC cells down-regulates MMP9 expression by suppressing GLI2-mediated GPNMB transcription. (discussion section, page 16-17, paragraph 2)

3. Even though tangeretin acts by downregulating GLI2 transcription factor, the molecular target of tangeretin is still unknown. I would encourage the authors to briefly comment about the upstream activators of GLI2 as the possible molecular target of tangeretin.

Response: According to the comment, we have added the discussion about the mechanism by which tangeretin downregulates GLI2 in the Revised manuscript as below.

In addition to the Hedgehog signaling, GLI2 has identified as a direct target of the TGF-β/SMAD pathway [13]. Previous work also uncovered the inhibition of Tan in TGF-β1 expression in human retinal pigment epithelial cells [36]. These conclusions suggest that Tan may reduce GLI2 expression by suppressing the TGF-β/SMAD pathway. Related field research is warranted in future work. (discussion section, page 15-16, paragraph 3)

4. Is there any weight loss noticed with the treatment group compared to the control group in vivo tumor xenograft studies?

Response: In this study, the generated xenograft tumors have little effect on the animal’s health and behavior. Owing to the light xenograft weight, there was no significant difference in body weight between the treatment group and the control group.

5. In the abstract, background section, the following sentence should be rewritten,

“However, the precise action of Tan in the development of esophageal squamous cell carcinoma (ESCC) has remained a mystery”

Response: We have revised the sentence to “However, the precise role of Tan in the development of esophageal squamous cell carcinoma (ESCC) remains unclear” in the Revised manuscript.

6. The manuscript will get benefit from additional English proof reading.

Response: Thank you for your valuable suggestion. According to the comment, we have re-revised the English in the Revised manuscript as below (for instance).

eg. Functional studies in various cancer cell lines have highlighted the tumor-inhibitory potential of Tan [8-10], but its precise action in ESCC pathogenesis remains unclear. (introduction section, page 12, paragraph 2)

eg. A loop graph was drawn for the visualization of the enrichment results. (method section, page 5, paragraph 2)

eg. We identified 587 genes with a significant variation following Tan treatment in TE-1 cells, in which 361 genes were down-regulated and 226 genes were up-regulated (S1 Table), and the cluster heat map of differentially expressed genes was shown in Fig 4A. (result 3 section, page 11, paragraph 2)

eg. Emerging evidence supports the notion that Tan exhibits potent anti-tumor efficacy against numerous human cancers [5,6]. (discussion section, page 14, paragraph 4)

Reviewer #3:

Remarks to the Author:

The article “The anti-tumor activity of tangeretin in esophageal squamous cell carcinoma by

inhibiting GLI2-mediated transcription of GPNMB’’ is an interesting study detecting the effect of tangeretin on migration and invasion in esophageal cancer cells by downregulating GLI2-mediated transcription.

However, the existing data are partially supportive of the authors' conclusions. Additional experiments are needed to fully support their findings. Minor revision is needed in recommending this manuscript for publication.

Minor issues:

1. In Fig.1A the author mentioned that the IC50 of tangeretin in TE-1 and KYSE150 is 85.92 and 94.34 μg/ml respectively, but why the author has selected the dose of 20 μg/ml concentration for evaluation of the anti-migratory effect tangeretin in these cells is unclear. Moreover, the IC50 value of tangeretin in normal esophageal cells need to be included in the result. The effect of tangeretin on normal cell growth and proliferation need to be included.

Response: Thank you for your kind recommendations. Based on the use of 1/5-1/2 IC50 of tangeretin in studies studying the effect of tangeretin on cancer cell behaviors, we used 20 μg/ml concentration (1/4-1/5) to assay the effect of tangeratin on ESCC cell migration and invasion in this study. In this study, we found that 20 μg/ml of tangeretin can significantly repress ESCC cell migration and invasion. When the concentration of tangeratin is high, the cell viability and migration will be significantly inhibited, which is not conducive to the study of the mechanism of tangeratin (rescue experiments). According to the comment, we have added the IC50 value of tangeretin in normal esophageal HET-1A cells to show the effect of tangeretin on cell growth in the S1 Fig and Revised manuscript as below.

Intriguingly, Tan (1-100 μg/ml) did not significantly affect the growth of normal esophageal HET-1A cells, and the IC50 value of Tan was 520.6 μg/ml against HET-1A cells (S1 Fig), suggesting the safety of Tan at low doses as an anti-tumor drug. (page 15, paragraph 2)

2. Authors mentioned that Gli2 downregulation by tangeretin causes reduction in cancer cell migration and invasion. Authors only showed the expression of MMP9 and PCNA, but Gli2 itself controls the transcription of EMT (epithelial-mesenchymal transition) specific molecules like Slug, Snail and also controls molecules like VEGF, Cyclin D1, Cyclin D2, Cyclin E. Hence, the effect of tangeretin on the expression these proteins should be determined by western blot analysis.

Response: Thank you for your valuable comments. According to the comment, we have added the discussion in the Revised manuscript as below.

Some important metastasis-related markers, such as EMT specific molecule Snail, angiogenesis-related molecule VEGF, and cell cycle regulator Cyclin D1, have been identified as target genes of GLI2 TF [39,40]. Our data also uncovered the repression of Tan in the expression of Slug, Snail, VEGF, and Cyclin D1 in ESCC cells (S2 Fig), suggesting that Tan may downregulate their expression via GLI2 and thus influences ESCC metastasis. (page 16, paragraph 2)

3. The resolution of Fig 3 (A-D), Fig. 5 A, B are very poor and should be changed with higher resolution figures.

Response: Thank you for your kind recommendations. We have provided higher resolution figures in the Revised Figure 3 and 5.

6. PLOS authors have the option to publish the peer review history of their article (what does this mean?). If published, this will include your full peer review and any attached files.

Do you want your identity to be public for this peer review? For information about this choice, including consent withdrawal, please see our Privacy Policy.

Reviewer #1: No

Reviewer #2: No

Reviewer #3: No

---

## [Decision Letter · Decision Letter 1]

17 Jul 2023

PONE-D-23-03144R1The anti-tumor activity of tangeretin in esophageal squamous cell carcinoma by inhibiting GLI2-mediated transcription of GPNMBPLOS ONE

Dear Dr. Yao,

Thank you for submitting your manuscript to PLOS ONE. After careful consideration, we feel that it has merit but does not fully meet PLOS ONE’s publication criteria as it currently stands. Therefore, we invite you to submit a revised version of the manuscript that addresses the points raised during the review process. The original reviewers reviewed the revised manuscript, and though they overall had positive recommendations, two of them suggested minor changes.

We look forward to receiving your revised manuscript.

Kind regards,

Arunava Roy, Ph.D.

Academic Editor

PLOS ONE

Journal Requirements:

Reviewers' comments:

Reviewer's Responses to Questions

**Comments to the Author**

1. If the authors have adequately addressed your comments raised in a previous round of review and you feel that this manuscript is now acceptable for publication, you may indicate that here to bypass the “Comments to the Author” section, enter your conflict of interest statement in the “Confidential to Editor” section, and submit your "Accept" recommendation.

Reviewer #1: All comments have been addressed

Reviewer #2: All comments have been addressed

Reviewer #3: (No Response)

2. Is the manuscript technically sound, and do the data support the conclusions?

Reviewer #1: Yes

Reviewer #2: Yes

Reviewer #3: Yes

3. Has the statistical analysis been performed appropriately and rigorously? 

Reviewer #1: Yes

Reviewer #2: Yes

Reviewer #3: Yes

4. Have the authors made all data underlying the findings in their manuscript fully available?

Reviewer #1: Yes

Reviewer #2: Yes

Reviewer #3: Yes

5. Is the manuscript presented in an intelligible fashion and written in standard English?

Reviewer #1: Yes

Reviewer #2: Yes

Reviewer #3: Yes

6. Review Comments to the Author

Reviewer #1: The revised manuscript entitled "The anti-tumor activity of tangeretin in esophageal squamous cell carcinoma by inhibiting GLI2-mediated transcription of GPNMB" is addressed all the concerns raised by reviewers.

The manuscript could be accepted after quality check especially the figures.

Reviewer #2: The authors made necessary changes in the manuscript. I recommend accepting the manuscript. However, the following typos should be modified to provide a clear message to the readers.

1. Furthermore, Tan is the capacity of enhancing the sensitivity of the drugs and reducing the toxicity induced by chemotherapy

2. A previous document proved the modulation of GPNMB in MMP9 in chronic obstructive pulmonary disease.

3. Therefore, while studying the anti-cancer mechanism of Tan, it is very important to study how overcomes these limitations for enhancing the therapeutic efficacy and reaching the clinical use

Reviewer #3: Remarks to the Author:

The article “The anti-tumor activity of tangeretin in esophageal squamous cell carcinoma by

inhibiting GLI2-mediated transcription of GPNMB’’ is an interesting study. Minor revision is needed in recommending this manuscript for publication.

Minor issues:

The Quality of the images should be increased.

1. In Fig.1A, 1B, 1C, 1D need to be replaced with high resolution images. All the cell images 1B, 1C need scale bars as many of them lack scale bars into it. In fig. 1A and the graphs in Fig. 1B, 1C and 1D the font size should be increased for clear visibility.

2. In Fig.2 A and 2B all the fluorescent images needed scale bars into it. The font size of the graphs in these figure need to be increased.

3. Fig. 3E needs to be changed with high resolution image.

4. Fig. 4A, 4B, 4C font size need to be increased.

5. Fig. 6A, 6B need to be needs to be changed with high resolution images with increased font size.

6. Font size of fig.7B needs to be increased.

7. Fig. 8C, 8D and 8E needs to be changed with high resolution images and scale bars must be added in each cell images.

8. Fig. 9A, 9B scale bars need to be added in each immunofluorescence cell images.

7. PLOS authors have the option to publish the peer review history of their article (what does this mean?). If published, this will include your full peer review and any attached files.

Reviewer #1: **Yes: **Jayaprakash Narayana Kolla

Reviewer #2: No

Reviewer #3: No

---

## [Author Response · Author response to Decision Letter 1]

28 Aug 2023

PONE-D-23-03144R1

The anti-tumor activity of tangeretin in esophageal squamous cell carcinoma by inhibiting GLI2-mediated transcription of GPNMB

PLOS ONE

Dear Dr. Yao,

Thank you for submitting your manuscript to PLOS ONE. After careful consideration, we feel that it has merit but does not fully meet PLOS ONE’s publication criteria as it currently stands. Therefore, we invite you to submit a revised version of the manuscript that addresses the points raised during the review process.

The original reviewers reviewed the revised manuscript, and though they overall had positive recommendations, two of them suggested minor changes.

We look forward to receiving your revised manuscript.

Kind regards,

Arunava Roy, Ph.D.

Academic Editor

PLOS ONE

Dear Editor,

We have revised the manuscript according to Reviewers’ comments and journal requirements. We look forward to hearing from you. Thank you very much.

Journal Requirements:

Response: We have checked the reference list and ensured that no retracted documents were cited in our manuscript. We have also revised the reference style according to the PLOS ONE's style requirements in the Revised manuscript.

Reviewers' comments:

Reviewer's Responses to Questions

Comments to the Author

1. If the authors have adequately addressed your comments raised in a previous round of review and you feel that this manuscript is now acceptable for publication, you may indicate that here to bypass the “Comments to the Author” section, enter your conflict of interest statement in the “Confidential to Editor” section, and submit your "Accept" recommendation.

Reviewer #1: All comments have been addressed

Reviewer #2: All comments have been addressed

Reviewer #3: (No Response)

2. Is the manuscript technically sound, and do the data support the conclusions?

Reviewer #1: Yes

Reviewer #2: Yes

Reviewer #3: Yes

3. Has the statistical analysis been performed appropriately and rigorously?

Reviewer #1: Yes

Reviewer #2: Yes

Reviewer #3: Yes

4. Have the authors made all data underlying the findings in their manuscript fully available?

Reviewer #1: Yes

Reviewer #2: Yes

Reviewer #3: Yes

5. Is the manuscript presented in an intelligible fashion and written in standard English?

Reviewer #1: Yes

Reviewer #2: Yes

Reviewer #3: Yes

6. Review Comments to the Author

Reviewer #1: The revised manuscript entitled "The anti-tumor activity of tangeretin in esophageal squamous cell carcinoma by inhibiting GLI2-mediated transcription of GPNMB" is addressed all the concerns raised by reviewers.

The manuscript could be accepted after quality check especially the figures.

Response: We have improved the quality of the manuscript including the figures in the Revised manuscript.

Reviewer #2: The authors made necessary changes in the manuscript. I recommend accepting the manuscript. However, the following typos should be modified to provide a clear message to the readers.

1. Furthermore, Tan is the capacity of enhancing the sensitivity of the drugs and reducing the toxicity induced by chemotherapy

Response: We have changed the sentence to “Furthermore, Tan enhances the sensitivity of the chemotherapy drugs and reduces chemotherapy-induced toxicity” in the Revised manuscript.

2. A previous document proved the modulation of GPNMB in MMP9 in chronic obstructive pulmonary disease.

Response: We have changed the sentence to “A previous document proved that GPNMB can influence MMP9 expression in chronic obstructive pulmonary disease” in the Revised manuscript.

3. Therefore, while studying the anti-cancer mechanism of Tan, it is very important to study how overcomes these limitations for enhancing the therapeutic efficacy and reaching the clinical use

Response: We have changed the sentence to “Therefore, it is very important to study how these limitations are resolved for their clinical use.” in the Revised manuscript.

Reviewer #3: Remarks to the Author:

The article “The anti-tumor activity of tangeretin in esophageal squamous cell carcinoma by inhibiting GLI2-mediated transcription of GPNMB’’ is an interesting study. Minor revision is needed in recommending this manuscript for publication.

Minor issues:

The Quality of the images should be increased.

1. In Fig.1A, 1B, 1C, 1D need to be replaced with high resolution images. All the cell images 1B, 1C need scale bars as many of them lack scale bars into it. In fig. 1A and the graphs in Fig. 1B, 1C and 1D the font size should be increased for clear visibility.

Response: According to the comment, we have provided high resolution images and increased the font size in the Revised Figure 1.

2. In Fig.2 A and 2B all the fluorescent images needed scale bars into it. The font size of the graphs in these figures need to be increased.

Response: We have added scale bars for all the fluorescent images and increased the font size in the Revised Figure 2.

3. Fig. 3E needs to be changed with high resolution image.

Response: We have provided higher resolution images in the Revised Figure 3.

4. Fig. 4A, 4B, 4C font size need to be increased.

Response: We have increased the font size in the Revised Figure 4.

5. Fig. 6A, 6B need to be needs to be changed with high resolution images with increased font size.

Response: We have changed with high resolution images with increased font size in the Revised Figure 6A and 6B.

6. Font size of fig.7B needs to be increased.

Response: We have added the font size in the Revised Figure 7B.

7. Fig. 8C, 8D and 8E needs to be changed with high resolution images and scale bars must be added in each cell images.

Response: We have changed with high resolution images and added scale bars for all microscopy images in the Revised Figure 8.

8. Fig. 9A, 9B scale bars need to be added in each immunofluorescence cell images.

Response: We have added scale bars in each immunofluorescence cell images in the Revised Figure 9.

7. PLOS authors have the option to publish the peer review history of their article (what does this mean?). If published, this will include your full peer review and any attached files.

Do you want your identity to be public for this peer review? For information about this choice, including consent withdrawal, please see our Privacy Policy.

Reviewer #1: Yes: Jayaprakash Narayana Kolla

Reviewer #2: No

Reviewer #3: No

---

## [Editor Report · Decision Letter 2]

1 Sep 2023

The anti-tumor activity of tangeretin in esophageal squamous cell carcinoma by inhibiting GLI2-mediated transcription of GPNMB

PONE-D-23-03144R2

Dear Dr. Yao,

We’re pleased to inform you that your manuscript has been judged scientifically suitable for publication and will be formally accepted for publication once it meets all outstanding technical requirements.

Kind regards,

Arunava Roy, Ph.D.

Academic Editor

PLOS ONE
---

## [Editor Report · Acceptance letter]

4 Jan 2024

PONE-D-23-03144R2 

PLOS ONE

Dear Dr. Yao, 

I'm pleased to inform you that your manuscript has been deemed suitable for publication in PLOS ONE. Congratulations! Your manuscript is now being handed over to our production team.

Kind regards, 

on behalf of

Dr. Arunava Roy 

Academic Editor

PLOS ONE